# Influence of Messa di Voce speed on vocal stability of professionally trained singers

**Marie Köberlein**[1]*, **Jonas Kirsch**[1], **Michael Döllinger**[2], **Matthias Echternach**[1]

**1** Division of Phoniatrics and Pediatric Audiology, Department of Otorhinolaryngology, LMU Munich University Hospital, Munich, Germany, **2** Division of Phoniatrics and Pediatric Audiology at the Department of Otorhinolaryngology Head & Neck Surgery, University Hospital Erlangen, Friedrich-Alexander-University Erlangen-Nürnberg, Erlangen, Germany

* marie.koeberlein@med.uni-muenchen.de (MK)

## Abstract

### Introduction

Messa di Voce (MdV) is a challenging task for singers, requiring an even modulation of sound pressure level (SPL) on a stable pitch. This study concentrates on the effects of fast or slow task-speed on voice stability parameters, and the associated laryngeal behavior. The focus is set on professionally trained singers.

### Material and methods

Ten professionally trained, healthy singers (5 female, 5 male) were asked to perform MdV exercises, i.e., a gradual increase and decrease of SPL, on the vowel [i:] on a stable fundamental frequency ($f_o \approx 247$ Hz for females and $f_o \approx 124$ Hz for males). First, each phase, i.e., increasing or decreasing SPL, should take 3 s. Second, each phase should take 1 s. The tasks were recorded by high-speed videolaryngoscopy (HSV), electroglottography, and audio signals. The following parameters were calculated and compared to the sound pressure level (SPL) curve: Electroglottographic (EGG) and Glottal Area Waveform (GAW) Open Quotients ($OQ_{EGG}$, $OQ_{GAW}$), Closing Quotient ($ClQ_{GAW}$) relative to start, Relative Average Perturbation ($RAP_{Audio/EGG/GAW}$), and Sample Entropy ($SE_{EGG}$).

### Results

In most subjects, no correlation of vibrato and SPL course was detected. Instabilities with higher $SE_{EGG}$ occurred at the start/end of the slow task, but not around the SPL apex. Generally, negative correlations of SPL to $OQ_{GAW}$, $ClQ_{GAW}$ and $RAP_{Audio}$ were present. $RAP_{EGG}$ and $RAP_{GAW}$ were not significant. In five subjects the decreasing phase of the slow task was 1–2 s longer. The majority of subjects ended the tasks softer than they had started.

---

**Data availability statement:** The data set's numerical files are available from https://doi.org/10.5281/zenodo.14900471.

**Funding:** The authors M. Echternach and M. Döllinger were supported by the Deutsche Forschungsgemeinschaft. Grant Numbers: Ec409/5-1/Deutsche Forschungsgemeinschaft Ec409/1-4/Deutsche Forschungsgemeinschaft DO1247/21-1/ Deutsche Forschungsgemeinschaft https://www.dfg.de/ The funders had no role in study design, data collection and analysis, decision to publish, or preparation of the manuscript. There was no additional funding received for this study.

**Competing interests:** The authors have declared that no competing interests exist.

## Conclusion

RAP values and SE suggest high laryngeal stability in professional singers. Vibrato did not play a role in the variation of SPL in the presented cohort. The data suggest that SPL variation is mainly controlled on vocal fold level rather than by vocal tract resonances.

## Introduction

In the training of western-classical singing, the Messa di Voce (MdV) exercise is estimated to be a big challenge [1]. Being capable of performing a well-balanced MdV is referred to as a sign of good vocal technique and thus, MdV is frequently an important element of singing education [1]. However, the complex physiological details of the exercise are not yet understood. In a MdV task, a pitch should be sustained while the intensity is evenly and symmetrically modulated from soft to loud and again back to soft. Herein, the difficulty is that the subglottic pressure ($p_{sub}$) impacts both basic voice characteristics, i.e., the intensity and the fundamental frequency ($f_o$) [2,3]. For the coordination of the maneuver where one parameter is altered and the other is constant, other components of voice control must be mastered simultaneously. For the modulation of intensity, resonance strategies in the vocal tract [4,5] and Maximum Flow Declination Rate (MFDR) – which could be altered by both, by $p_{sub}$ and glottal resistance – and Maximum Area Declination Rate (MADR) on laryngeal level [6–9] can be used. At the same time, the vocal folds (VF) configuration must be adjusted to keep a stable $f_o$, and the breathing apparatus has to maintain a suitable $p_{sub}$ in spite of a decreasing lung volume over time [10,11]. Several studies have explored aspects of MdV. It has been observed that with improving MdV control, the sound pressure level (SPL) increased, and vibrato was more present, leading to the assumption that these two parameters might be linked [12,13]. A symmetry of the SPL course was rarely found, but convex or concave shaped courses and different phase lengths, suggesting deviations of perception and quantitative measures [14–16]. Regarding the vocal tract's influence on spectral intensity, the SPL course correlated to lip opening, jaw opening, pharynx width, uvula elevation, and vertical larynx position [17]. Another little researched aspect of MdV is the speed of the execution. For a faster task, the breath dosage for $p_{sub}$ is presumed to be less difficult, but vibrato would not have enough time to evolve. A previous study regarding speed of MdV explored if untrained subjects would show breaking points, instabilities or other behaviors standing out as a baseline [18]. Contrary to the expectations, the study did not find any instabilities or breaking points, but negative correlations between SPL and Open Quotients, Closing Quotient and Relative Average Perturbation (RAP).

The presented follow up study deals with the topic of fast or slow MdV execution speeds in professionally trained singers of western classical style regarding voice stability parameters and laryngeal behavior. The study further aims to compare the results to those of untrained singers. The outcomes could provide information for different fields of voice training on which speed of the MdV task should be applied

for different purposes and how it could be mastered. It has been hypothesized that trained singers compensate for speed differences and will show constant behaviors independent of speed as well as more even and/or symmetric SPL curves, but less vibrato in the fast task.

## Materials and methods

This prospective observational cohort-study follows the STROBE guidelines. After approval from the local ethical committee (Medical Ethics Committee of the University of Munich, 20–282) and obtaining informed written consent by the participants, 10 adult, vocally healthy, western classical professionally trained singers were included in the study. Their classification following the Bunch-Chapman taxonomy [19] is documented in Table 1. The participants were recruited from the local university of music community through personal contact. The prerequisite was at least a bachelor's degree and vocal health, which was verified by medical history, laryngoscopy and Dysphonia Severity Index (DSI) assessment.

The subjects performed two MdV tasks on the vowel [i:], modulating intensity in two connected phases: 1) From *piano* (soft phonation) towards *forte* (loud phonation), denoted with the musical term *crescendo*; 2) Back from *forte* to *piano*, denoted with the musical term *decrescendo*. In the first task (*slow*), each phase should last 3 s, and in the second task (*fast*), each phase should take 1 s. The vowel [i] was chosen, since the associated tongue position with tongue elevation and forward shift provides the best visibility on the vocal folds. Female subjects phonated at pitch B3 ($f_o \approx 247$ Hz), and male subjects phonated at pitch B2 ($f_o \approx 124$ Hz). Each subject was provided with the respective pitch played on a piano right before each task. The $f_o$ were chosen in the region of the speaking voice, since here the vocal apparatus works most efficiently and the probability of other disruptive factors' influence, e.g., higher tension for higher pitches, would be minimized. The duration of the task was not controlled but self-monitored by the subjects. This limitation was accepted in the study design because an acoustic metronome would have disturbed the audio recordings, and the application of a visual metronome was impossible due to the examiner standing right in front of the participants and the space-consuming technical equipment around them.

As in previous studies [21,22], transnasal endoscopic highspeed videos (HSV) were recorded with a Photron Fastcam SA-X2 [23] with an ENF GP Fiberscope [24] and a Storz light source [25] (20k fps, image resolution of 386 × 320 pixels, monochromatic). Simultaneously, an electroglottographic (EGG) (Glottal Enterprises EG2-PCX2 [26]) and an audio signal (DPA 4061 [27], 4 cm distance to the corner of the mouth) were captured using a NI USB-6251 BNC [28] (20 kHz). The audio signal's amplitude was calibrated using a sound level meter (Tecpel DSL 331 [29]) and the Sopran software [30]. The HSV sequences were post-processed by means of

**Table 1. The subjects' voice classification following the Bunch-Chapman taxonomy [19] and DSI [20].**

| Subject | Voice Type | Taxonomy | | DSI |
|---|---|---|---|---|
| 1 | Baritone | 7.1\|4.5 | Fulltime voice student university–postgraduate\|Regional touring | 5.0 |
| 2 | Baritone | 4.5 | Regional touring | 5.6 |
| 3 | Soprano | 7.1\|4.5 | Fulltime voice student university–postgraduate\|Regional touring | 6.6 |
| 4 | Mezzosoprano | 7.1\|4.5 | Fulltime voice student university–postgraduate\|Regional touring | 5.2 |
| 5 | Baritone | 7.1\|4.5 | Fulltime voice student university–postgraduate\|Regional touring | 7.2 |
| 6 | Baritone | 4.5 | Regional touring | 6.4 |
| 7 | Tenor | 7.1\|4.5 | Fulltime voice student university–postgraduate\|Regional touring | 7.5 |
| 8 | Mezzosoprano | 7.1\|4.5 | Fulltime voice student university–postgraduate\|Regional touring | 2.8 |
| 9 | Soprano | 4.5 | Regional touring | 6.6 |
| 10 | Soprano | 7.1\|4.5 | Fulltime voice student university–postgraduate\|Regional touring | 3.8 |

Fast-Fourier-Transformation, rotation and cropping, as described before [31]. The segmentation and calculation of the glottal area waveform (GAW) and phonovibrograms (PVG) [32] from the images were performed with the Glottis Analysis Tools (GAT) software [33] (Version 2020). Each signal was separated into non-overlapping windows of 100 ms. By using the software Multi Signal Analyser (MSA) [34], for each window the following averaged parameters were calculated: Open quotients from GAW ($OQ_{GAW}$, open threshold of 5% of the cycle's maximum) and electroglottography ($OQ_{EGG}$, Howard criterion [35]), closing quotient from GAW ($CIQ_{GAW}$), Relative Average Perturbation values from all signals (RAP, version 1 [36]), SPL and the EGG's absolute sample entropy (SE, based on the first two Fourier descriptors (corresponding to $f_o$ and $2*f_o$) used for registration events) [37,38] (Table 2). Vibrato parameters were extracted from the GAW signal using the Praat Software (University of Amsterdam, Netherlands). The $f_o$ extraction process involved a cross-correlation-based two-pass operation, where an algorithm adjusted the floor and ceiling $f_o$ values. During the first pass (50–700 Hz), $f_o$ was extracted, and its values were calculated. The first and third quartiles of the $f_o$ distribution (q1, q3) were determined, and new optimal values for the $f_o$ floor and ceiling were computed using the formulas: $floor = 0.7 \cdot q1$ and $ceiling = 1.5 \cdot q3$. These values were then used to perform the second pass of $f_o$ extraction. To analyze the modulation of $f_o$ over time and reconstruct the resulting minima and maxima from GAW cycles, extrema recognition was applied using Matlab's islocalmin and islocalmax functions [39]. Minima and maxima were identified with a minimal time distance of 1/10 s between them. Relevant time regions in $f_o$ were then selected based on a minimum prominence threshold of 0.5 semitones. The prominence of a local extremum indicates how distinct it is relative to surrounding peaks, measured by the vertical distance between the peak and its neighboring valley.

For statistical analysis, Spearman rank correlation test was used with two-sided hypothesis testing (testing the alternative hypothesis that the correlation is not 0) with a significance level of =0.05. For further details see [18].

Since the dynamic parts of the task were in focus of this study, potentially stationary SPL peaks were excluded by analyzing only until 95% of the maximum-to-minimum SPL range. To enable direct parameter comparison during *crescendo/decrescendo*, the time courses were normalized: timepoint 0 describes the start of the task, 1 is the point in time in which the SPL reaches its maximum, and timepoint 2 the end of the task. Since *crescendo* and *decrescendo* regions were chosen according to a 0–95% minimum-to-maximum threshold, for visual clarity, each region has only a temporal length of 0.95 instead of 1 (although 5% of amplitude *crescendo* and *decrescendo* does not necessarily mean that 5% of the time has passed). The ranges of *crescendo* and *decrescendo* were calculated separately.

An n-way Analysis of Variance (ANOVA, Matlab function anovan() [41], was used for the examination of the statistical relationship between measured parameters and slow/fast task or *crescendo/decrescendo* phase. Each parameter value is described by subject number, sex, task speed, rising or falling SPL, and time since start of the task. The elapsed time and subject number are handled as random variables to compensate for correlations due to proximity in time or an origin in

**Table 2. Signal sources and analyzed parameters. Parameter calculation was conducted according to Schlegel [40].**

| GAW | EGG | Audio |
|---|---|---|
| Closing Quotient ($CIQ_{GAW}$) | | |
| Open Quotient ($OQ_{GAW}$) | Open Quotient ($OQ_{EGG}$, Howard criterion [35]) | |
| | Absolute Sample Entropy (SE) | |
| Relative Average Perturbation ($RAP_{GAW}$) | $RAP_{EGG}$ | $RAP_{Audio}$ |
| | | Sound Pressure Level (SPL) |
| | Fundamental Frequency ($f_o$) | |

same subject measurement. The impacts of the other factors are analyzed individually and in combination with a significance level of $\alpha = 0.05$, and p-values are compensated using the Tukey-Kramer procedure [42–44] due to pairwise comparisons and Bonferroni's method [45] to compensate for the number of analyzed parameters per subject (see Matlab's multcompare() [46]). Only those factors/interactions are analyzed post-hoc via pairwise comparisons, which are reported as significant by the ANOVA ($\alpha = 0.05$).

## Results

All ten subjects completed both MdV tasks on a stationary pitch (average $f_o$ for slow and fast task in male subjects: 121±5 Hz and 120±6 Hz; average $f_o$ for slow and fast task in female subjects: 245±9 Hz and 244±8 Hz, see S1 Fig). The subjects took on average 5.72 seconds for the slow task (Max: 8.48 s, Min: 2.98 s, StD: 1.65 s) and 1.72 seconds for the fast task (Max: 2.68 s, Min: 1.11 s, StD: 0.52 s). Fig 1 shows the SPL courses of all subjects and both tasks. During both tasks, the overall median SPL extent was 18.5 dB(A) (StD slow: 3.57 dB(A); StD fast: 4.01 dB(A)). The SPL apex positions differed between subjects and tasks and the majority of subjects ended the task with lower SPL compared to the beginning.

The durations of active *crescendo/decrescendo* phases were mostly equal. In five subjects (subjects 2, 3, 8, 9, 10), the *decrescendo* of the slow task was 1–2 seconds longer than the *crescendo* (Fig 2).

The SE showed increases in the beginning or end of the slow task and in the end of the fast task (Fig 3). Only in one subject (subject 6), an increased SE peak was found near the SPL apex in the slow task, while this occurred for 3 subjects during the fast task (subjects 2, 4, 6). Subject 6 showed higher SE before and after the apex in both tasks, but not in the beginnings and ends. For subject 2, the SE was generally increased in the fast task.

Regarding the relationship of SPL and $OQ_{GAW}$ or CIQ, there were significant negative correlations in both slow and fast tasks (Fig 4). The outcome for $OQ_{EGG}$ was ambiguous.

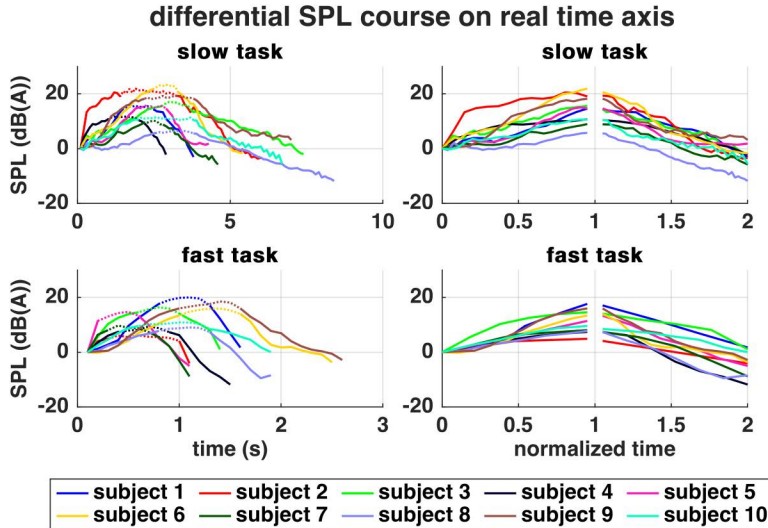

**Fig 1. Courses of SPL in reference to initial SPL.** In the left column, the SPL apex with the 95-100% criterion is marked by the dotted line. The right column shows the normalized time course with 1 representing the SPL apex and 2 the end of the task. Note that 0 to 1 and 1 to 2 does not necessarily represent a similar duration.

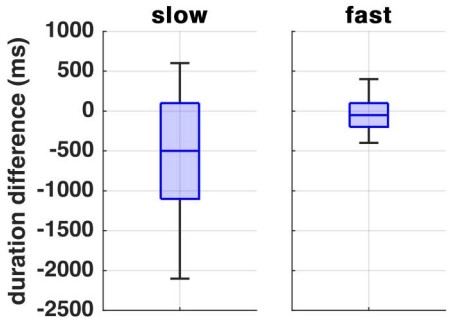

**Fig 2. Duration differences of SPL phases calculated by crescendo minus decrescendo with the slow task on the left and the fast task on the right side.**

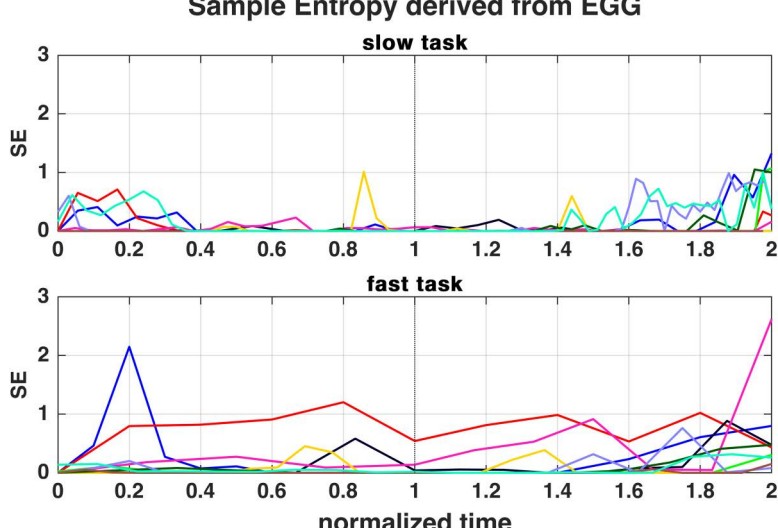

**Fig 3. Sample Entropy derived from Electroglottography (EGG) for time-normalized slow (upper diagram) and fast task (lower diagram).** For the color assignment of the subjects see Fig 1.

$RAP_{Audio}$ showed significant correlations to SPL in both slow and fast tasks (Fig 5), more pronounced in the decrescendo phase of the slow task.

Phonovibrograms of each 5 glottal cycles were calculated for the start (at 100 ms), the SPL apex window, and the end (100 ms before end) of each task, Fig 6. Here, for many subjects, i.e., subjects 3, 6, 8 and 9, the end of the phonation was associated with less glottal closure. At the apex, most subjects show either a longer closed phase (black parts) or larger oscillation amplitude (brighter red).

As shown in Fig 7, only subjects 1, 6, and 7 showed vibrato courses correlated contrariwise to the SPL course during the slow task. In the fast task this occurred only for subject 1. For the others there was no clear connection between SPL course and vibrato.

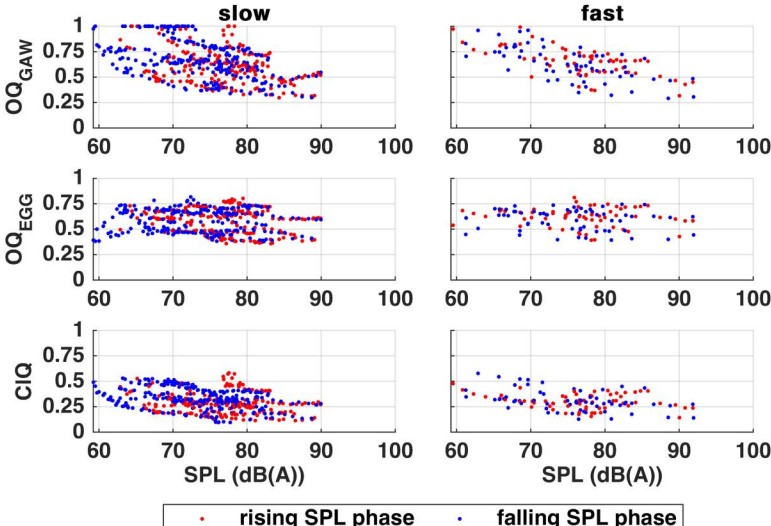

**Fig 4. Open Quotients (OQ) of Glottal Area Waveform (GAW) and Electroglottography (EGG) and GAW-derived Closing Quotient (CIQ) in relation to Sound Pressure Level (SPL) of slow (left) and fast (right) tasks.** Red dots mark crescendo and blue dots mark decrescendo data.

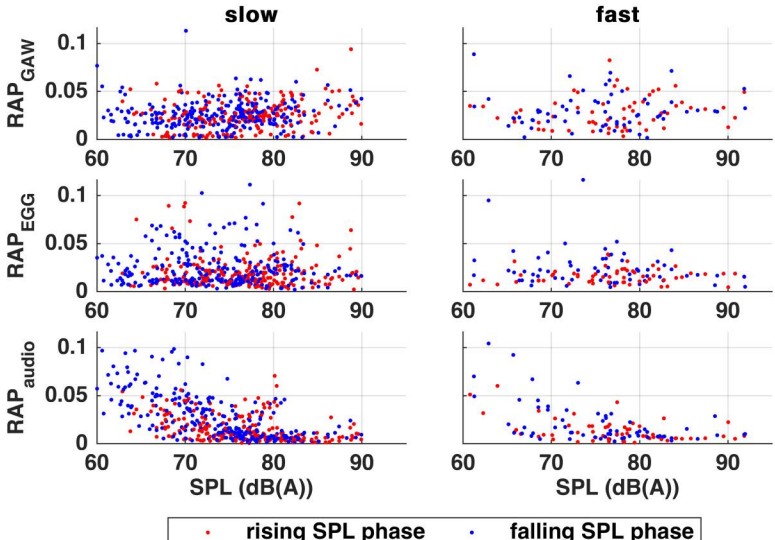

**Fig 5. Relative Average Perturbation (RAP) of Glottal Area Waveform (GAW), Electroglottography (EGG) and Audio signals in relation to Sound Pressure Level (SPL) of slow (left) and fast (right) tasks.** Red dots mark crescendo and blue dots mark decrescendo data.

| Subj. | Task | Start | Apex | End |
|-------|------|-------|------|-----|
| 1 ♂ | slow | | | |
| | fast | | | |
| 2 ♂ | slow | | | |
| | fast | | | |
| 3 ♀ | slow | | | |
| | fast | | | |
| 4 ♀ | slow | | | |
| | fast | | | |
| 5 ♂ | slow | | | |
| | fast | | | |
| 6 ♂ | slow | | | |
| | fast | | | |
| 7 ♂ | slow | | | |
| | fast | | | |
| 8 ♀ | slow | | | |
| | fast | | | |
| 9 ♀ | slow | | | |
| | fast | | | |
| 10 ♀ | slow | | | |
| | fast | | | |

**Fig 6. Phonovibrograms [32] of each 5 glottal cycles from the start (at 100 ms), the SPL apex window, and the end (100 ms before end) of each task.**



**Fig 7. All subjects' courses of SPL (black line) in comparison to vibrato speed (Hz, blue line) and vibrato extent (semitones, dashed blue line) of slow and fast tasks.**

## Discussion

This investigation focused on vocal fold oscillatory characteristics and voice stability parameters derived from audio, EGG and GAW signals when executing MdV exercises at different speeds in professionally trained singers. It was expected that the professionally trained singers would exhibit more stability compared to the untrained singers of a previous study [18] and show smaller differences regarding speed. Furthermore, it was expected that vibrato would correlate to SPL at least in the slow task [12,13]. According to these expectations, there was no major difference between the two execution speeds. A difference in comparison to untrained singers was a higher stability in laryngeal configuration, represented by $RAP_{GAW}$ and $RAP_{EGG}$. Contrasting with the expectations, no consistent positive correlation of vibrato and SPL was detected in the tasks performed by professionally trained subjects. Instead, in cases where vibrato occurred, some negative correlations occurred and/or vibrato stopped during the SPL apex.

In general, there are three general approaches to control SPL and loudness: modification of the transglottic pressure difference, mainly controlled by $p_{sub}$ [47], modification of glottal resistance by modification of vocal fold adduction, and modification of the vocal tract [17,48–50]. Thus, for professional Western classical singing, the MdV represents a challenging task. This is due to the fact that $p_{sub}$ not only influences SPL but also $f_o$ which should be kept stable throughout the MdV [49,51]. Furthermore, in order to decrease both, impact and shear stress to the VFs, singers train to use a reduction of the glottic resistance in order to increase the transglottic air pulse and thus increase the maximum flow declination rate, the so-called flow phonation [49]. Also, singers train to adjust vocal tract resonances to voice source partials in order to increase SPL and/or loudness. Consecutively, it could be expected for professional singers and in contrast to untrained voices that the rise of SPL during MdV could be associated with less control of SPL by $p_{sub}$ but by flow phonation and/or voice source/vocal tract resonance adjustments. However, similar to the untrained singers and in accordance with existing literature, negative correlations were found for the $CIQ_{GAW}$ and $OQ_{GAW}$ of both tasks with regard to SPL [8,18,52–54], recall Fig 4. Such negative correlation should reflect a control of SPL on the VF level through higher MADR [9], but unlikely a control by voice source/vocal tract resonance adjustments. Further, the negative correlation for $OQ_{GAW}$ and SPL contradicts the expectation of a small rise of $OQ_{GAW}$ during increase of SPL due to a small abduction of the VFs for a flow phonation. However, in contrast to $OQ_{GAW}$, only about half of the subjects showed such correlation in $OQ_{EGG}$ of both tasks. This can be explained by the different signal sources of EGG and GAW. Since the one-dimensional GAW represents the two-dimensional view onto the glottis from above, it captures the definite closing of the glottis in each cycle, which is crucial for the final push of the air pulse and is thus closely connected to SPL. EGG on the other hand represents the vertical contact area of the VFs and is thus suitable for a detection of $f_o$ and VF mass, which is rather associated with vocal registers. Since the tasks were performed in the comfortable range of speaking voice, the VF mass does not seem to have played a major role in SPL modulation. The question arises whether this behavior would be the same in other regions of the subjects' $f_o$-range.

Similar to the untrained singers, some of the professional singers ended the task softer than they had started, recall Fig 1. It could be speculated if this is due to phonation threshold pressure which might need a higher pressure to start the self-sustained VF oscillation, than to keep an already oscillating system going at lower pressures. However, also a miscalculation of the subjects could be the reason, and as demonstrated by Dejonckere et al. [55], visual SPL feedback might have led to a more symmetrical execution [55] Also in this context, it was checked in the subjects who exhibited a 1–2s longer decrescendo than crescendo, if the phases would have been equal at the point where they reached the initial SPL. However, this assumption has not been confirmed. Another possible explanation for the differences in phase length could be a difference of the subjects' perceived SPL apex and the calculated 95% apex phase. In subject 2, for example, the calculation of an apex was generally difficult due to a long SPL plateau, which was longer than the active *crescendo*/*decrescendo* phases (recall Fig 1). The PVG show less glottal closure at the end of both tasks which might hint towards a reduction of SPL using more airflow and less collision forces. Regardless of execution speed, this technique also appears in female singers in the beginning of the tasks.

Regarding the shape of the SPL curve, the singers frequently exhibited steep SPL courses with rather long SPL-apex-plateaus (recall Fig 1). Last is in accordance to observations by Titze et al. [16]. Yet, the different types of SPL curve shapes as shown by Collyer et al. [15], i.e., concave or convex configuration, were not clearly detectable.



In contrast to the untrained singers who had shown correlating variations of all RAP signals, the professionals' RAP values showed negative correlations to SPL only in the audio signals of both tasks (slow in 9/10 subjects, fast in 7/10) suggesting high $f_o$ stability, recall Fig 5. The significant correlations occurred in *crescendo* as well as *decrescendo* phases. It is noticeable that there were – in contrast to $OQ_{GAW}$, $OQ_{EGG}$ and ClQ – some individual differences and outliers. However, as can be seen in Fig 5, even such outliers did not exceed the value of.12 by a great amount and are therefore in the lower physiological range.

In untrained singers, raised SE values before the apex of the fast task had been reported [18]. Such was not present in the professional singers in the presented study who showed higher SE values, i.e., more irregularities, rather at the start/end of the slow task, recall Fig 3. Since professional singers are trained to fill concert halls with their voices, they might phonate more stably at higher pressure levels. Also, the SPL extent in professional singers was at the same microphone distance higher than in untrained singers (medians of 18 dB(A) compared to 11–13 dB(A)). The PVG at the apex of both tasks showed longer closed phases or larger oscillation amplitudes (Fig 6), which agrees with the results of $OQ_{GAW}$ being smaller for higher SPL. It seems likely that the professional singers use the VF configuration to vary SPL in MdV tasks.

Most of the subjects did not show vibrato courses correlating to SPL (Fig 7). This might be due to the experimental circumstance where subjects tried to sing a clearly defined $f_o$ on exact length in order to only change one variable, i.e., SPL. It is unknown how often these subjects practice MdV exercises in their daily life, how long each phase would take usually and how they would perform it in artistical contexts. However, for the few cases where vibrato parameters correlated to SPL, the correlation was negative, which is in contrast to the expectation that bigger vibrato supports higher SPL. The reason for this outcome remains unclear.

## Limitations

In the study setup, no optical metronome was provided which would have improved standardization of the tasks' length. Furthermore, MdV is usually described as a slow task to make air dosage and stability more challenging. Thus, it is possible that the length of the slow task was even too fast to make the challenges of MdV occur. However, due to the high-speed camera's technical properties, the recording time could not be extended without reducing spatial or temporal resolution. Furthermore, due to the extensive study setup, only a small number of participants could be measured, and thus the statistical power of the data is limited.

## Practical implications

The observations that the untrained singers showed correlations of SPL and RAP, and struggled before the apex of the fast task, while the professionals did not, leads to the assumption that as prerequisite for a MdV, one must master stable phonation on one pitch at the different levels of SPL with a good vocal fold closure. These can afterwards be linked smoothly in slow MdV tasks which provide more time for coordination. Finally, the execution speed can be increased. While slow MdV poses a challenge on breath control, fast MdV needs faster laryngeal and vocal tract coordination. Thus, for advanced training, different task speeds could be applied with regard to the training's focus.

## Conclusion

Regarding the execution speed of MdV exercises in professionally trained singers, no differences were observed, but negative correlations of SPL to OQ, $ClQ_{GAW}$ and $RAP_{Audio}$ were present in both tasks. No breaking points were detected. In contrast to untrained singers, $RAP_{EGG}$ and $RAP_{GAW}$ were not significant with regard to the speed, suggesting a higher laryngeal stability in professional singers. Contrary to untrained singers, Sample Entropy was raised rather at start/end of the slow task but not around the SPL apex. Vibrato did not play a significant role in the variation of SPL but longer closed phases and higher oscillation amplitudes served for an SPL increase. For a further decrease at the end of the tasks, the

professional singers reduced VF closure resulting in less glottal collision. Women also used this technique in the beginning of the tasks.

## Supporting information

**S1 Fig. Courses of fundamental frequency $f_o$ on phase-wise normalized time.** The SPL apex of the MdV is represented by time point 1, the end of the task is represented by 2. Note that 0–1 and 1–2 do not necessarily represent similar durations.
(TIF)

## Author contributions

**Conceptualization:** Marie Köberlein, Matthias Echternach.

**Data curation:** Marie Köberlein.

**Formal analysis:** Marie Köberlein, Jonas Kirsch.

**Funding acquisition:** Michael Döllinger, Matthias Echternach.

**Investigation:** Marie Köberlein, Matthias Echternach.

**Methodology:** Jonas Kirsch, Matthias Echternach.

**Project administration:** Marie Köberlein.

**Software:** Jonas Kirsch, Michael Döllinger.

**Supervision:** Matthias Echternach.

**Visualization:** Jonas Kirsch.

**Writing – original draft:** Marie Köberlein, Matthias Echternach.

**Writing – review & editing:** Marie Köberlein, Jonas Kirsch, Michael Döllinger, Matthias Echternach.

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
