## [Decision Letter · Decision Letter 0]

17 Apr 2025

PONE-D-25-10020Influence of Messa di Voce speed on vocal stability of professionally trained singersPLOS ONE

Dear Dr. Köberlein,

Thank you for submitting your manuscript to PLOS ONE. After careful consideration, we feel that it has merit but does not fully meet PLOS ONE’s publication criteria as it currently stands. Therefore, we invite you to submit a revised version of the manuscript that addresses the points raised during the review process. 

In particular, I trust that the points raised will improve manuscript's readability. Regarding Reviewer 1's comment on the figure quality, I belive the poor impression was due to the poor compression applied by the PLOS ONE management system. The original high-quality TIFF files were avilable for download, and I personally confirmed that they are in good condition. However, I would still recommend resizing the small fonts (e.g., in Fig. 7) as they might be illegable in the typeset paper, even at double-column width.

We look forward to receiving your revised manuscript.

Kind regards,

Seung-Goo Kim, Ph.D.

Academic Editor

PLOS ONE

Journal Requirements:

https://pubmed.ncbi.nlm.nih.gov/39883711/

In your revision ensure you cite all your sources (including your own works), and quote or rephrase any duplicated text outside the methods section. Further consideration is dependent on these concerns being addressed.

Reviewers' comments:

Reviewer's Responses to Questions

**Comments to the Author**

1. Is the manuscript technically sound, and do the data support the conclusions?

Reviewer #1: Partly

Reviewer #2: Yes

2. Has the statistical analysis been performed appropriately and rigorously? 

Reviewer #1: I Don't Know

Reviewer #2: I Don't Know

3. Have the authors made all data underlying the findings in their manuscript fully available?

Reviewer #1: No

Reviewer #2: Yes

4. Is the manuscript presented in an intelligible fashion and written in standard English?

Reviewer #1: Yes

Reviewer #2: Yes

5. Review Comments to the Author

Reviewer #1: This study investigates the Messa di Voce (MdV) in professional singers, a task requiring even modulation of sound pressure level (SPL) on a stable pitch. Specifically, the authors examine the effects of speed (slow and fast) on voice stability parameters and laryngeal behavior. The study involved 10 professionally trained singers (5 male and 5 female) performing Messa di Voce (MdV) exercises, with a gradual increase and decrease of SPL at a stable fundamental frequency (247 Hz for females and 124 Hz for males). To analyze the singers' performance, the authors utilized high-speed video laryngoscopy, electroglottography (EGG), and acoustic recordings of singing voices. Several parameters were extracted from the EGG signals, including glottal area waveform, open glottis, closing quotient, relative average perturbation (RAP), and sample entropy.

The study found that vibrato had no correlation with SPL, and instabilities with higher sample entropy based on EGG were primarily observed at the start and end of the slow task, but not around the SPL effects. Additionally, most subjects ended the task with a softer sound than they had started. The results suggest that relative average perturbation and sample entropy indicate high laryngeal stability in professional singers, while vibrato did not play a role in SPL variation in this cohort.

Strengths of the Study

Interesting and Important Topic: The study addresses a challenging and underexplored aspect of vocal modulation in professional singers.

Comprehensive Methodology: The authors employed multiple modalities—high-speed video laryngoscopy, EGG, and acoustic analysis—to capture detailed insights into laryngeal behavior and SPL modulation.

Clear Experimental Setup: The distinction between slow and fast tasks provides an effective framework for evaluating speed-related effects on Messa di Voce (MdV).

Areas for Improvement

As the task requires singers to maintain a stable pitch, rhythm, and intensity. However, the study does not include contour plots for fundamental frequency, rhythm, and intensity, which are crucial to verifying the stability of these parameters. Instead, the authors only plot sound pressure level (SPL) over time for the slow and fast tasks. The authors should plot these parameters to confirm that they remained stable throughout the task, as this is a key assumption in their analysis.

The introduction mentions MFDR as an important indicator, yet the study relies solely on EGG-based measurements.

Suggestion: It would be beneficial to extract voice source parameters such as MFDR, closing quotient, and opening quotient from the singing voice itself rather than exclusively from EGG. This would provide a more comprehensive analysis of how voice source parameters impact Messa di Voce (MdV).

Quality of Figures: The figures, especially the last few, are of poor quality, making them difficult to read. The authors should improve the resolution and clarity of the figures to enhance readability and ensure that critical details are not lost.

Integration of Figures with Text: There is a lack of coherence between the figures and the text. Some figures are not well-referenced in the discussion, making it difficult to follow the analysis. The authors should directly reference figures within the text and provide a clearer explanation of their relevance to the discussion.

Scientific Backing for Findings:

The authors should provide stronger scientific explanations for why they obtained their results. For example, why does sample entropy indicate higher laryngeal stability? Why is vibrato not correlated with SPL in this cohort?

Suggestion: A more thorough discussion of underlying physiological mechanisms and previous literature would strengthen the scientific validity of the conclusions.

The scatter plots presented in the later sections should be examined more critically. Some outliers and spurious values are present in SPL, RAP, sample entropy, and other measures.

The authors should analyze whether these values result from individual variations among singers or measurement inconsistencies. Investigating potential correlations among different parameters could also provide deeper insights into Messa di Voce (MdV) mechanisms.

This study explores an interesting and relevant topic in voice science, providing valuable insights into Messa di Voce (MdV) and voice stability in professional singers. However, several areas require improvement before publication. The lack of fundamental contour plots, limited use of MFDR from singing voices, poor figure quality, and insufficient scientific backing for findings weaken the study’s overall impact. Addressing these concerns and integrating the suggested improvements would enhance the clarity, coherence, and robustness of the study.

Reviewer #2: Thank you for the opportunity to read and review this manuscript. The topic is interesting and relevant. The text is well written and provides essential information for understanding messa di voce as a vocal exercise or a test to assess singers' vocal production. I have made some critical suggestions for the final version of the manuscript.

ABSTRACT

The abstract is well structured and provides relevant information that is consistent with the research objective.

However, the way the research objective is described is vague. For example, I cannot understand whether what the authors want to investigate is the “effect” of task speed on some other variable. The authors' writing of the objective does not make it clear that this association is being investigated. Therefore, I suggest that they review the way they present the objective. The aim seems to be much clearer in the title of the work and could be transposed (with the necessary adjustments in the writing) to the abstract and to the end of the introduction (where the objective is described).

INTRODUCTION

The authors provided an adequate contextualization of the topic. The most relevant studies and research hypotheses were presented.

However, what problem would be solved if this hypothesis were confirmed is unclear. In this sense, the authors should present in the introduction the context or problem that will be solved with this knowledge. Although the study's rationale is well presented, its relevance is unclear to the reader.

METHOD

The authors did not define the study design. It is not clear how the singers were recruited nor what the exclusion criteria would be for participants.

The authors need to provide more information to justify some choices. For example, please explain why the vowel [i] was chosen and not another vowel. It may seem obvious, but it is not so for all potential readers of this manuscript.

A detailed description of the data collection procedures is missing, which would allow the study to be reproducible. For example, how was the duration of the task controlled? Was it self-monitored? What is the justification for choosing phonation at these frequencies?

Was the HSV, voice, and EGG data collected simultaneously or sequentially?

What justifies the use of Spearman's correlation? Were the data non-normally distributed? Was the sample size small? While they used Spearman's correlation (a non-parametric test), the authors used ANOVA (a parametric test). I would like to hear the authors' justifications for this.

RESULTS

The results were presented in an appropriate and didactic manner. The authors reviewed the hypotheses and discussed the results found.

I recommend that the authors comment on the practical implications of the findings of this study for vocal training, whether through the intervention of singing teachers or SLPs.

CONCLUSION

The conclusions presented are compatible with the study design, the proposed objectives, and the results found.

6. PLOS authors have the option to publish the peer review history of their article (what does this mean? ). If published, this will include your full peer review and any attached files.

**Do you want your identity to be public for this peer review?** For information about this choice, including consent withdrawal, please see our Privacy Policy .

Reviewer #1: No

Reviewer #2: **Yes: ** Leonardo Lopes

---

## [Author Response · Author response to Decision Letter 1]

30 Apr 2025

Dear Editor and Reviewers,

Thank you for your positive and helpful assessment. In the document "Response to reviewers" we address every issue raised.

Kind regards,

Marie Köberlein and colleagues

---

## [Decision Letter · Decision Letter 1]

12 May 2025

Influence of Messa di Voce speed on vocal stability of professionally trained singers

PONE-D-25-10020R1

Dear Dr. Köberlein,

We’re pleased to inform you that your manuscript has been judged scientifically suitable for publication and will be formally accepted for publication once it meets all outstanding technical requirements.

Kind regards,

Seung-Goo Kim, Ph.D.

Academic Editor

PLOS ONE

Additional Editor Comments (optional):

Reviewers' comments:

Reviewer's Responses to Questions

**Comments to the Author**

1. If the authors have adequately addressed your comments raised in a previous round of review and you feel that this manuscript is now acceptable for publication, you may indicate that here to bypass the “Comments to the Author” section, enter your conflict of interest statement in the “Confidential to Editor” section, and submit your "Accept" recommendation.

Reviewer #1: All comments have been addressed

Reviewer #2: All comments have been addressed

2. Is the manuscript technically sound, and do the data support the conclusions?

Reviewer #1: Partly

Reviewer #2: Yes

3. Has the statistical analysis been performed appropriately and rigorously? 

Reviewer #1: I Don't Know

Reviewer #2: Yes

4. Have the authors made all data underlying the findings in their manuscript fully available?

Reviewer #1: No

Reviewer #2: Yes

5. Is the manuscript presented in an intelligible fashion and written in standard English?

Reviewer #1: Yes

Reviewer #2: Yes

6. Review Comments to the Author

Reviewer #1: The authors have addressed the previous comments. The manuscript is now in good shape and can be considered for acceptance.

Reviewer #2: Congratulations once again on the manuscript and thank you for all the answers to the questions I asked. I have no additional comments and I understand that the manuscript is ready for publication.

7. PLOS authors have the option to publish the peer review history of their article (what does this mean? ). If published, this will include your full peer review and any attached files.

**Do you want your identity to be public for this peer review?** For information about this choice, including consent withdrawal, please see our Privacy Policy .

Reviewer #1: No

Reviewer #2: **Yes: ** Leonardo Lopes

---

## [Editor Report · Acceptance letter]

PONE-D-25-10020R1

PLOS ONE

Dear Dr. Köberlein,

I'm pleased to inform you that your manuscript has been deemed suitable for publication in PLOS ONE. Congratulations! Your manuscript is now being handed over to our production team.

Kind regards,

on behalf of

Dr. Seung-Goo Kim

Academic Editor

PLOS ONE